Review article

# Challenges and recent advancements in the synthesis of α,α-disubstituted α-amino acids

Yu Zhang [1,2] ✉, Jaro Vanderghinste[2], Jinxin Wang[1] & Shoubhik Das [2,3] ✉

α,α-Disubstituted α-amino acids (α-AAs) have improved properties compared to other types of amino acids. They serve as modifiers of peptide conformation and as precursors of bioactive compounds. Therefore, it has been a long-standing goal to construct this highly valuable scaffold efficiently in organic synthesis and drug discovery. However, access to α,α-disubstituted α-AAs is highly challenging and largely unexplored due to their steric constraints. To overcome these, remarkable advances have been made in the last decades. Emerging strategies such as synergistic enantioselective catalysis, visible-light-mediated photocatalysis, metal-free methodologies and $CO_2$ fixation offer new avenues to access the challenging synthesis of α,α-disubstituted α-AAs and continuously bring additional contributions to this field. This review article aims to provide an overview of the recent advancements since 2015 and discuss existing challenges for the synthesis of α,α-disubstituted α-AAs and their derivatives.

Amino acids (AAs) are fundamental building blocks in biology and approximately 500 AAs have been identified in nature[1]. Their name stems from the fact that all of them contain at least one amine and one carboxylic acid moiety, making them attractive building blocks in organic synthesis. Additionally, the presence of these functional groups results in a relatively high hydrophilicity, a property that is currently exploited for the enhancement of drug delivery. Currently, about 40-60% of new drug molecules exhibit poor water solubility, and therefore face serious bioavailability issues. In this respect, the use of AAs has been proven to improve the biopharmaceutical properties of drugs such as permeability, stability, and solubility[2]. Along the same line, α,α-disubstituted α-AAs are non-proteinogenic AAs that, next to the amino and carboxylic acid moiety, bear two substituents on the α-carbon atom (Fig. 1a). This type of skeleton is present in many natural products and is therefore of interest to the pharmaceutical industry as a component of drug candidates[3–5]. However, the construction of such quaternary centers, i.e. carbon atoms with four distinct non-hydrogen substituents, remains a challenge in organic synthesis, primarily because the steric hindrance mandates a multi-step approach, which must not affect the

reactive amino and carboxyl groups[6,7]. Considering their importance, numerous methods have been developed to construct α,α-disubstituted α-AAs over the past decades (Fig. 1b). However, there are some obvious limitations for these typical methods. For instance, a traditional strategy to obtain these scaffolds, the Strecker reaction, was described in the 1850s[8–10]. With this simple and efficient method, α,α-disubstituted α-AAs were afforded via the hydrocyanation of the activated imine derivatives and subsequent hydrolysis of the resulting amino nitriles. Later, the enantioselective Strecker reaction has also been developed to synthesize chiral α,α-disubstituted α-AAs. However, this strategy suffered from low yields, the use of toxic reagents such as cyanides, and undesired enolization of the corresponding ketimines. The challenge in employing the Strecker reaction for the synthesis of α,α-disubstituted α-AAs also stems from the lower electrophilicity of the iminyl carbon of ketimines compared to aldimines. Afterwards, the coupling of enolates of Schiff-base-derived α-amino esters with diverse electrophiles has been reported for the formation of α,α-disubstituted α-amino acids[11,12]. The difference in electrophilicity of the iminyl carbon also indirectly influences these types of reactions, as the reaction site, either anionic

[1]Shanghai Frontiers Science Center for Chinese Medicine Chemical Biology, Institute of Interdisciplinary Integrative Medicine Research, Shanghai University of Traditional Chinese Medicine, No. 1200, Cailun Road, 201203 Shanghai, China. [2]Department of Chemistry, University of Antwerp, Groenenborgerlaan 171, 2020 Antwerpen, Belgium. [3]Department of Chemistry, University of Bayreuth, Bayreuth, Germany. ✉e-mail: yzhang@shutcm.edu.cn; Shoubhik.Das@uni-bayreuth.de

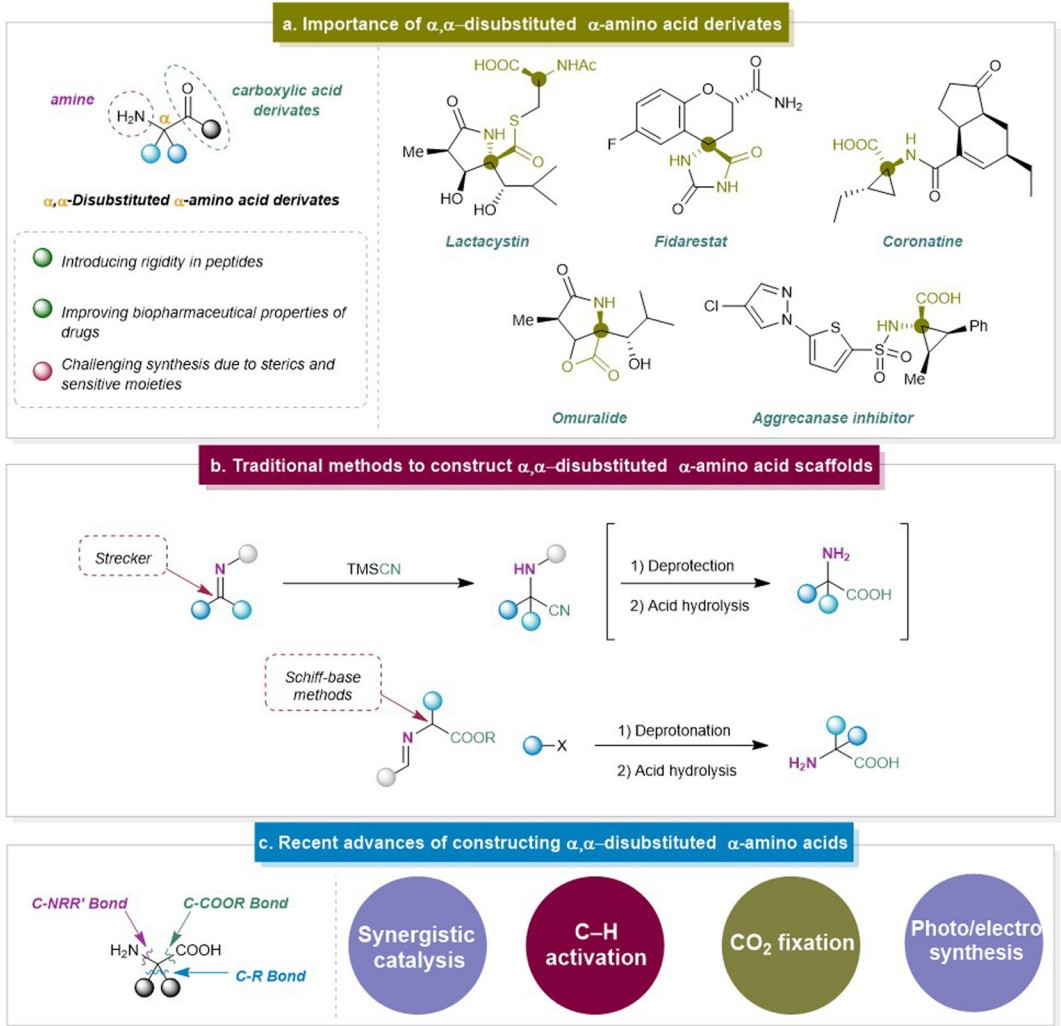

**Fig. 1 | General overview on α,α-disubstituted α-AAs. a** Importance of α,α-disubstituted α-amino acid derivates. **b** Traditional methods to construct α,α-disubstituted α-amino acid scaffolds. **c** Recent advances of constructing α,α-disubstituted α-AAs.

or radical in nature, is in direct conjugation with the imine. Therefore, these intermediates will have improved stabilization in aldimines and ketimines, respectively, due to the amount of electron-rich phenyl substituents[13]. The foundation of phase-transfer catalysis with achiral Schiff base esters, however, was laid by O'Donnell and Ghosez to afford dialkylated α-amino acids[14–16]. This method allowed mild conditions and the ability to perform reactions on a large scale, however, the prior assembly of Schiff base esters was required.

Considerable attention has been devoted to developing methodologies for the synthesis of α,α-disubstituted α-AAs. In the last decade, the emerging strategies including the metal-free C-H activation[17], organocatalysis[18], $CO_2$ fixation[19], and photo(electro) catalysis[20] brought new opportunities to access α,α-disubstituted α-AAs. Moreover, there are some advances based on the modification of typical methods such as replacing hazardous reagents with environmentally friendly chemicals and developing other catalytic systems to access α,α-disubstituted α-AAs efficiently. Even though several reviews have already elaborated by using typical methods[21–24], we deem it necessary to introduce and summarize the recent surge of advancements and updates of α,α-disubstituted α-AAs synthesis. Therefore, we will focus on the recent contributions to the synthesis of α,α-disubstituted α-AAs since 2015, as the latest review on this topic was then published by the Kozlowski group[21]. In order to keep a clear overview, this review will be divided based on the diverse bond

formation that is being formed to provide the α,α-disubstituted α-amino acid derivative, which is either via the C-N bond formation, and the C-C bond formation (Fig. 1c).

## Conceptual overview for the synthesis of α,α-disubstituted α-AAs

To clearly illustrate the differences between the formation of C-N and C-C bonds, an overview of the key mechanistic principles is shown in Fig. 2. When performing the C-N bond disconnection, the challenges of this strategy immediately arise as both the α-carbon and nitrogen atom generally act as nucleophiles. In the first case, the inherent nucleophilic character of the α-carbon atom is maintained through the formation of an enamine intermediate, which attacks the nitrogen atom in electrophilic reagents such as nitrosoarenes or azodicarboxylates. Enantioselectivity in these procedures generally stems from steric hindrance in the enamine intermediate, by selecting an appropriate chiral secondary amine. However, it is also possible for the N-electrophiles to undergo radical coupling with an α-centered radical intermediate generated from the carbonyl substrate. When the negative charge is preserved on the nitrogen atom, the electrophilicity of the α-carbon is increased through coordination with transition metals. The ligands surrounding the metal center are carefully selected to enhance the nucleophilic attack at the reaction site, as well as block the nucleophile from certain angles to increase the enantioselectivity.

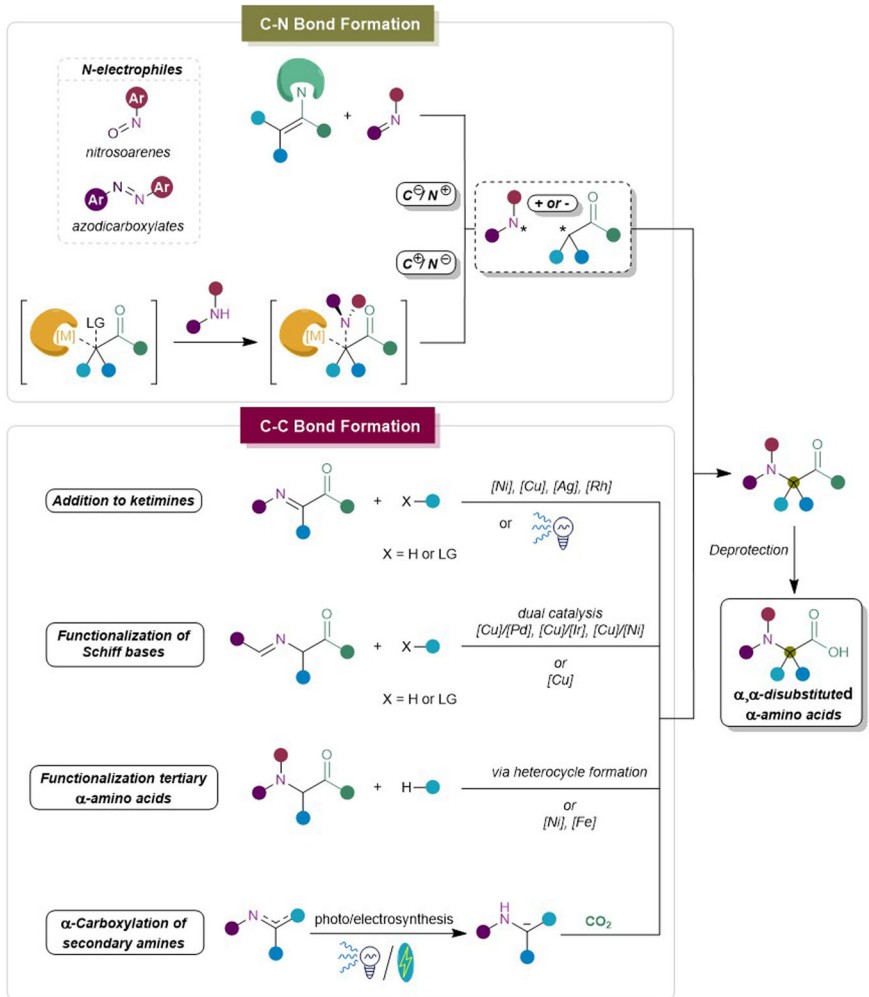

**Fig. 2 | Conceptual overview on α,α-disubstituted α-AAs.** Schematic representation of the mentioned strategies for the synthesis of α,α-disubstituted α-AAs classified by the type of bond formation.

The C-C bond formation includes diverse strategies with a well-established example being the addition at ketimines. One improvement since 2015 is the addition of previously challenging building blocks such as electron-rich alkenes via transition metal catalysis[25]. Moreover, ketimine Mannich reactions with diverse nucleophiles have been also further developed to induce enantioselectivity by using metal catalysts[26] with chiral organocatalysts or ligands[27]. In addition, multi-component coupling is another robust approach in this field to reach structural diversity[28]. Particularly, the metal-free visible-light-mediated methods towards α,α-disubstituted α-amino esters have been recently described to show the potential of emerging toolbox in the synthesis of α,α-disubstituted α-AAs[29].

Among the advances of constructing α,α-disubstituted α-AAs via C-C bond formation, one of the conceptual advances is using the synergistic catalysis via Cu/Pd, Ni/Cu and other cooperative systems to achieve the enantioselectivity which was challenging before[30–35]. Another conceptual improvement is to extend the coupling partners towards general alkyl halides[12] and versatile tertiary hydrocarbons which were highly challenging in previous reports[36]. The key to success is the radical-radical coupling of two radical species that are obtained through a one-electron process. α-Functionalization of general α-AAs was also reported recently to realize various α,α-disubstituted α-AAs, especially extending to the arylation and alkenylation of α-amino acids via multi-step pathways[37–39]. The key step of the C-C bond formation is the rearrangement reaction to realize the arylation of the amino acid α-

center in a diastereoselective manner. Besides the abovementioned indirect α-functionalization methods, direct α-C-H bond functionalization of α-amino acids is also becoming an emerging tool to construct the C-C bond. The key to achieving such an ideal transformation is to activate inert $C(sp^3)$-H bonds via a coordinating activation strategy[40]. At last, hydrocarboxylation of amines or imines to form the α,α-disubstituted α-AAs is also emerging recently. A different strategy is the construction of an α-amino carbanion via photoredox approaches to capture $CO_2$, thereby facilitating the synthesis of α,α-disubstituted α-AAs[41]. Moreover, visible-light-promoted 1,2-acyl migration was also applied to the synthesis of α,α-disubstituted α-AAs successfully. All these approaches will be exemplified and discussed critically in next sections.

## Synthesis of α,α-disubstituted α-AAs via C-N bond formation

Even though many methods such as the Gabriel synthesis[42], Buchwald-Hartwig reaction[43], or reductive amination[44,45] have been established for the C-N bond formation, the direct α-amination to provide α,α-disubstituted α-amino acids is impeded by the presence of untargeted electron-deficient reaction sites. To avoid interference with these groups, a straightforward route is to select a nitrogen-source that itself is electrophilic in nature. For this purpose, azodicarboxylates and nitrosoarenes have been well-described aminating agents due to the involvement of an

electron-accepting heteroatom adjacent to the nitrogen atom in the C-N bond formation.

## Electrophilic amination reactions to synthesize α,α-disubstituted α-amino acids

Advantage of using electrophilic aminating agents stems from the deep understanding of the nucleophilic character of the α-carbon in the substrate, which can be accessed through an enamine intermediate. Therefore, the C-N bond disconnection delivers two synthons, namely an electron-rich carbon and electron-poor nitrogen atom. A disadvantage of this technique, however, is the limitation in nitrogen reagents that can be utilized, as the nitrogen center needs to be turned electrophilic. Additionally, to obtain the unmasked amine, generally, a reductive N-N or N-O bond cleavage is performed using SmI$_2$ or Zn after the reaction[46,47]. Furthermore, asymmetric electrophilic amination has also been established via the formation of a chiral enamine that can sterically hinder one enantiomer by attacking the nitrogen electrophile in a selective manner[48,49]. However, this strategy is only feasible if a sufficiently electron-deficient carbonyl moiety is present to react with the chiral secondary amine.

In 2015, the group of de Alaniz provided an elegant approach for the synthesis of α,α-disubstituted α-amino carbonyl scaffolds containing sterically hindered aniline moieties to improve the metabolic stability and lipophilicity of drug molecules (Fig. 3a)[46,50,51]. By means of a Cu(I) catalyst, the reduction of α-bromo carbonyl substrate **1** yielded carbon-centered radical intermediate **I** that engaged in radical addition with a nitrosoarene electrophile **II**. In order to close the copper catalytic cycle, *N*-aryl hydroxylamine **2** was added as a precursor to the *N*-electrophile, as it yielded the nitrosoarene through single electron oxidation by the Cu(II) metal center and brought the catalyst back to its +1 oxidation state. Following the radical addition, persistent nitroxyl radical **III** was generated which underwent a radical-radical coupling reaction with **I** to form **IV**. Finally, a reductive N-O bond cleavage in the presence of SmI$_2$ delivered the α-aminated α,α-disubstituted carbonyl product **3**. The synthetic utility of their protocol was demonstrated by synthesizing carfentanil derivative **4** and cathepsin K inhibitor precursor **5**. Followed by the synthetic protocol of de Alaniz Later, the Ooi group also introduced a hydroxylamine as the *N*-electrophile[52]. In this case, the electrophile was activated by trichloroacetonitrile through the formation of an *O*-imino intermediate to make it a potent electrophile. This excellent approach elevated the state-of-the-art by achieving high enantioselectivity (up to 99% *ee*) owing to the addition of chiral 1,2,3-triazolium salt catalyst **6** (Fig. 3b). Using their strategy, the authors also performed intramolecular cyclization reactions to construct tetrahydroquinoxaline and isatin scaffolds which are valuable to synthesize anti-HIV agents[53] and biologically active alkaloids[54].

Inspired by the abovementioned strategies, in 2018, Feng and co-workers performed the organocatalytic hydroxyamination of α-substituted α-cyanoacetates with nitrosobenzene[55]. Previously, while excellent reactivity was achieved, a strong background reaction hampered enantioselectivity (22-59% *ee*), which became apparent when the Jørgensen group achieved nearly the full conversion at −78 °C within 5 h[56]. The Feng group, however, was able to considerably improve the enantioselectivity up to 96% *ee* by switching the Jørgensen's alkaloid catalyst by a chiral *N,N'*-dioxide/Mg(OTf)$_2$ complex. After the initial assessment of the ligand, the authors observed the beneficial effect of decreasing the number of bridging carbon atoms in the *N,N'*-dioxide scaffold. When decreasing this number from three to two, the value of *ee* increased from 56 to 75%. Adding an extra isopropyl substituent on the ligand delivered the chiral product in 85% excess, which, after further optimizations, was raised to 93% *ee*[57].

More recently, the Mohanan group employed the Takemoto chiral thiourea catalyst for the electrophilic amination of α-methylmalonamates with nitrosoarenes to construct *N*-alkyl-*N*-arylhydroxylamines[47]. Through bifunctional hydrogen bonding, the catalyst activated both the nitrosoarene and the malonamate substrate which induced enantioselectivity. While moderate to high enantioselectivity (50–91% *ee*) was achieved for most of the substrates, four examples exhibited enantioselectivity <50%. Among these four, two of them were the only examples that did not contain the model nitrosobenzene moiety, revealing the intolerance of the reaction protocol towards modifications in the nitrosoarene. However, during the Zn-mediated reductive N-O bond cleavage to provide the unmasked amine, the *ee* value dropped from 90% to 49%, exposing the challenge in the post-functionalization of the electrophilic amination.

Later, Chinchilla and co-workers developed a solvent-free procedure for the metal-free asymmetric α-amination of α,α-disubstituted aldehydes[58]. A carbamate-protected derivative of cyclohexa-1,2-diamine was used for this purpose which had already been used by the same group in their enantioselective Michael addition of α,α-disubstituted aldehydes to maleimides[59]. In their most recent work, the Chinchilla group targeted the C-N bond formation by selecting *N*-electrophilic azodicarboxylates. Remarkably, high yields and enantioselectivities were obtained for α-alkyl-α-aryl and α,α-dialkyl aldehydes, which were proven to be challenging in prior reports[60,61]. The key to achieving high conversion and enantioselectivity was to work in solvent-free conditions, hereby reaching full conversion in 24 h instead of 48 h. To further understand the selective formation of the *R* isomer as the major product in the reaction, extensive DFT calculations were performed which revealed four plausible transition states involving the stable *E* enamine intermediate (Fig. 3c). Between those, there were two main competing factors, the first one being the steric advantage of the azodicarboxylate engaging from the *Re* face (upper side), providing the *S* enantiomer (**TS2** vs. **TS4**). Opposite to this, there was the presence of an intermolecular hydrogen bond between one of the carbonyl moieties of the azodicarboxylate and the carbamate -NH of the catalyst, which was stronger and easier to form with the electrophile on the *Si* face (bottom side), leading to the *R* enantiomer (**TS1** vs. **TS3**). By calculating the energies of these transition states, it revealed that the stabilization due to hydrogen bonding surmounted the steric destabilization, which was reflected by **TS1** having the lowest energy and thereby clarified the presence of the *R* enantiomer as the major product. Additionally, an oxazolidinone scaffold was constructed without loss of chirality, via consecutive aldehyde reduction and esterification. These oxazolidinones are ubiquitous in natural products and pharmaceutical drugs, as they appear to be potent agents against resistant Gram-positive pathogenic bacteria strains[62,63].

Overall, some remarkable contributions have been added to the state of the art, as the construction of chiral α,α-disubstituted α-amino acids could be constructed with impeccable enantioselectivity. However, several challenges still remain, such as the post-functionalization of the N-O- or N-N-bearing reaction products towards the free amino acid, which is generally either inefficient or not reported. Additionally, the asymmetric electrophilic C-N bond formation for α,α-dialkyl substrates still remains elusive due to their inability to coordinate with a metal center or engage in π-π bonding, hereby sterically pushing the reaction towards one enantiomer.

## Nucleophilic amination reactions to synthesize α,α-disubstituted α-amino acids

Since the α-carbon to a carbonyl-containing group is generally nucleophilic in nature due to the acidity of the proton, additional measures are required to swap the polarity of the α-carbon so that an *N*-nucleophile can engage. In current methods, the activation of the carbon center generally occurs via coordination with a transition metal complex by decreasing the electron density on the carbon, thereby leaving it vulnerable for nucleophilic attack. At the same time, careful selection of the ligands can induce enantioselectivity and provide the α-functionalized products in impressive enantiomeric excess. However, the nucleophilic α-nitrogenation of tertiary carbon atoms

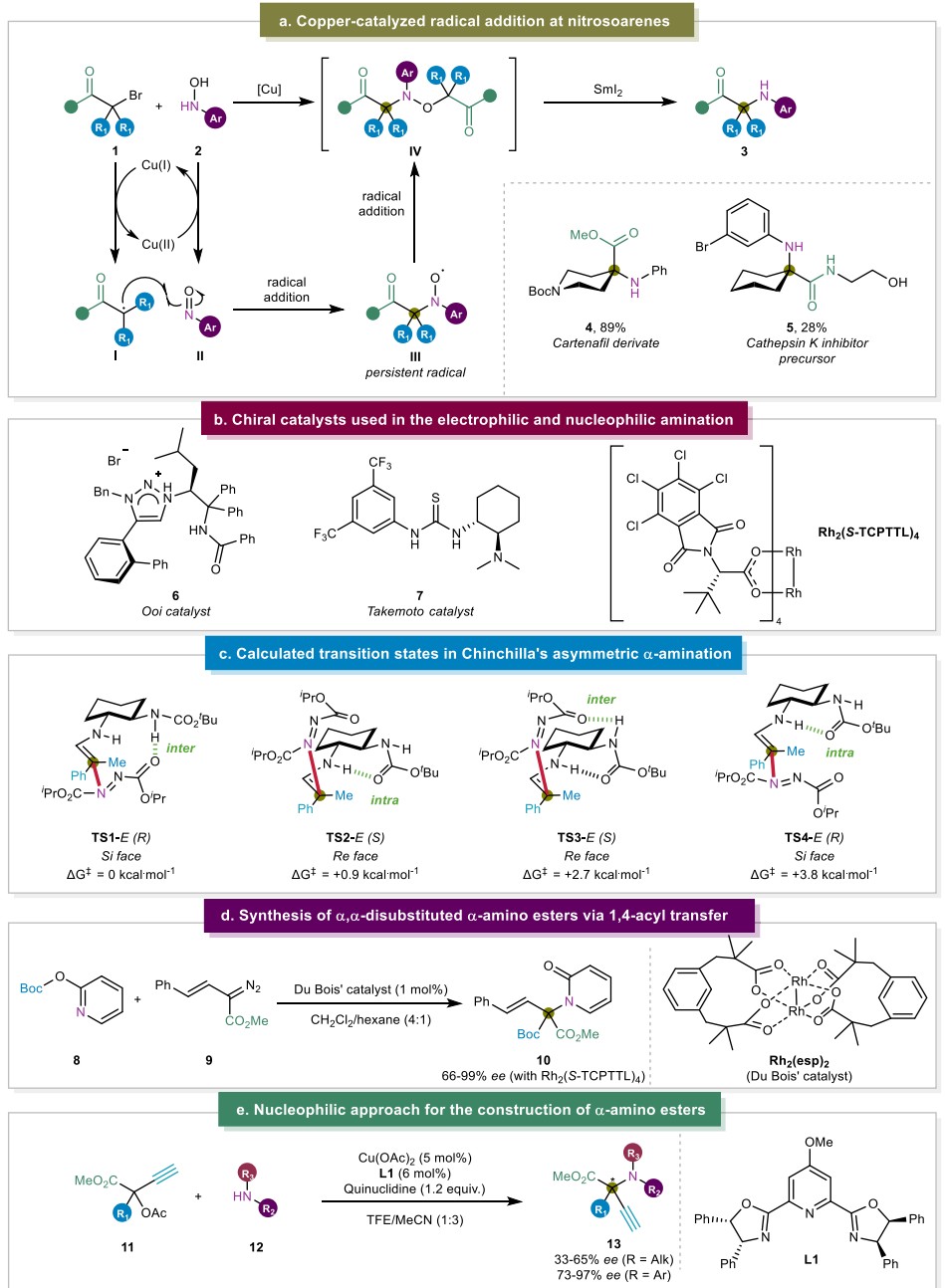

**Fig. 3 | C-N Bond Formation. a** Copper-catalyzed radical addition at nitrosoarenes[46]. **b** Chiral catalysts used in the electrophilic and nucleophilic amination[52]. **c** Calculated transition states in Chinchilla's solvent-free procedure[58].

**d** Synthesis of α,α-disubstituted α-amino esters via 1,4-acyl transfer[64].
**e** Nucleophilic approach for the construction of α,α-disubstituted α-amino esters[67].

remains relatively underexplored due to the difficulty in steric access to the reaction center, and the additional lack in stereocontrol.

The research group of Sun disclosed the rhodium-catalyzed dearomatisation of *O*-substituted pyridines to attain *N*-substituted 2-pyridones where nitrogen was introduced to construct a α,α-disubstituted center (Fig. 3d)[64]. Since *N*-substituted 2-pyridones are key scaffolds in pharmaceuticals and natural products, it is crucial that diverse routes to construct them in enantioselective manners are continuously explored[65,66]. In the case of the Sun group, quantum chemical calculations indicated the formation of a Rh-carbene intermediate that underwent a nucleophilic attack from the pyridine nitrogen atom and then left a pyridinium ylide upon recovery of the rhodium catalyst. Consecutively, an 1,4-acyl migration from *O*- to *C*-

center was triggered by the coordination of the Boc-moiety of the pyridine with the carbanionic center. Intrigued by the mechanistic information, a broad substrate scope was explored using the achiral Rh₂(esp)₂ (Du Bois' catalyst, esp = α,α,α',α'-tetramethyl-1,3-benzene-dipropionic acid), and the racemates of the functionalized pyridones **10** were achieved in moderate to high yields (51–90%). However, by swapping the achiral esp ligand of the complex to chiral *S*-TCPTTL (tetrachloro-*N*-phthaloyl-*tert*-leucinate) (Fig. 3b), the authors established the asymmetric variation of their strategy with up to 99% ee. The explanation for this enriched enantioselectivity was revealed by DFT calculations, which revealed that the preference of the procedure towards the *S* enantiomer was determined during the nucleophilic addition of the pyridine to the α,α-disubstituted center, as the Rh-

carbene adopted an α,α,α,α-chiral crown conformation that underwent strong π-π and CH-π interactions, depending on the shape of the pocket formed by the four ligands.

In 2022, Guo and co-workers found a strategy that allowed the asymmetric α-amination of propargylic esters with readily available amines as nucleophiles (Fig. 3e)[67]. The reason for introducing the alkyne moiety in the substrate was to facilitate the use of a chiral copper catalyst, as these are omnipresent in the enantioselective preparation of α,α-disubstituted centers in propargylic substrates. The α,α-disubstituted carbon center became prone for nucleophilic attack through activation via sequential copper coordination and deprotonation at the terminal alkyne. This strategy had already been reported by the Kleij group for the synthesis of γ-amino acids and tertiary sulfones, both bearing chiral α,α-disubstituted centers[68,69]. However, the synthesis of α-amino acids with this strategy was much more challenging, since the ester moiety on the electrophilic reaction center resulted in destabilization and therefore caused decomposition before it underwent a nucleophilic attack. Fortunately, Guo et al. managed to overcome this by introducing a methoxy substituent in the backbone of the catalyst ligand, thereby stabilizing the Cu-allenylidene intermediate. When exploring the substrate scope, the authors only observed moderate to high enantioselectivity (73–97% *ee*) when there was an aryl substituent on the α-carbon, as opposed to an alkyl substituent (33–65% *ee*). The authors attributed the latter results to steric hindrance as substituting a 2-napthyl for a methyl group reduced the enantioselectivity from 92% to 39%. In order to support their theory, a bulkier isopropyl substituent was introduced, which increased the *ee* to 65%. To demonstrate the application potential of this strategy, a 'click' cyclization was successfully performed (81%, dr >20:1) between α-aminated propargylic ester **13** and zidovudine, an azide-containing antiretroviral drug used to prevent and treat aids. Additionally, two bioactive molecules containing a free amine moiety were assessed as *N*-nucleophile, and provided the α,α-disubstituted α-amino ester product in 54% and 95% yield. Notably, the prepared α-amino esters did not lose chirality when used in the follow up peptide synthesis, as *dr* values of over 20:1 were obtained in both of the cases. With their procedure, the authors were able to reduce the traditional 11-step synthesis of α-ethylnorvaline to only 3 reaction steps[70], which clearly demonstrates the practicality of this method in both synthetic and pharmaceutical chemistry.

In summary, an advantage of the nucleophilic approach compared to the electrophilic amination is its lack for the need for reductive N-O or N-N bond cleavage, since the nucleophilic nature of the nitrogen atom is maintained, and no distinct reagents are required. However, the nucleophilic amination for the asymmetric synthesis of α,α-disubstituted α-amino acid derivatives still remains underexplored.

## Synthesis of α,α-disubstituted amino acids via the C-C bond formation

C-C bond formation provides a direct and efficient route to access α,α-disubstituted amino acids. Diverse strategies including the addition at ketimines, α-functionalization of amino acid Schiff bases, and α-functionalization of general α-amino acids have been developed. Even though these previous approaches are already well-established, they still contain some challenges and limitations. For instance, the introduction of bulky groups or aryl moieties at the α-carbon is a challenging task. Since 2015, some advancements have overcome these difficulties. Herein, the recent advances in constructing α,α-disubstituted α-AAs via C-C bond formation will be critically discussed.

### Recent advances of addition at ketimines

Along this direction, a notable example is the Strecker amino acid synthesis[8]. In this context, Shimizu et al. demonstrated a tandem *N*-alkylation-*C*-allylation of ketiminoesters using allyltributyltin and organoaluminium compounds[71]. Even though the imine nitrogen is evidently a challenging electrophile, this strategy provided an elegant route to functionalize this nitrogen atom. Herein, several examples published after 2015 will be discussed critically, mentioning associated challenges during the synthesis.

In 2015, Jia and co-workers developed a Ni(II)-catalyzed strategy for the enantioselective addition of styrenes to cyclic *N*-sulfonyl α-ketiminoesters (Fig. 4a)[25]. Notably, even though the addition of electron-deficient alkenes was well known, the direct addition of styrenes remained elusive before this[72]. With this procedure, the authors provided a feasible synthetic route for cyclic sulfonamides **16** (sultams) which are well known to have biological activities and have been used as medicinal compounds[73]. With this strategy in hand, both allylic and homoallylic products were obtained in excellent enantiomeric excesses (88 to >99% *ee*), depending on whether styrene or α-methylstyrene was used, respectively. Even though Jia et al. have been able to achieve an excellent scope of this reaction, mechanistic studies were still lacking in this work. In 2011, Shibasaki was able to construct chiral and cyclic α,β-diamino esters, where stereodivergent access was obtained through alteration of the catalyst metal center. Applications of Sr(O-iPr)$_2$ as the catalyst provided *anti* adducts in 84−97% *ee* and 17:83-4:96 d.r. (*syn/anti*), and using Bu$_2$Mg as the catalyst generated *syn* adducts in 80−95% *ee* and 90:10−93:7 d.r. (*syn/anti*)[74]. Inspired by this work, the Dixon group disclosed a silver-catalyzed enantioselective ketimine Mannich reaction of α-substituted isocyanoacetate esters **17**, thereby affording α,α-disubstituted α-AAs bearing two fully substituted carbon centers (Fig. 4b)[26]. The nitrogen atom of the (*E*)-configured ketimine is hypothesized to form a pivotal hydrogen bond with the amide N-H group. This interaction was used to orient both electrophiles and enabling the activation for the subsequent nucleophilic addition. The key to the success was that the aromatic ring of the ketimine was spatially distant from quinuclidine and quinoline units. Moreover, C-H/π-interactions with the aromatic ring of the carboxamide made additional binding as well as organization, which was responsible for the high enantioselectivity observed with this catalyst. After the enantio- and diastereodetermining C-C bond formation, subsequent ring closure and protonolysis lead to the formation of the N-DPP protected imidazoline **20** exhibiting the observed stereochemical configuration. In 2016, the Terada group also managed to obtain excellent stereocontrol in a Mannich-type reaction of α-iminophenylacetate esters using thionolactone nucleophiles[27]. For this, they introduced a chiral bis(guanidino)iminophosphorane as organocatalyst to reach diastereomeric ratios of up to 99:1. Despite the generation of vicinal α,α-disubstituted stereogenic centers, the pronucleophiles were limited to specific α-iminophenyl acetate esters with thionolactones.

Another strategy for the enantioselective synthesis of α-amino esters was disclosed by Ohshima and his team through a rhodium-catalyzed alkynylation of α-ketiminoesters where they showed that less acidic alkynes served as a proton source in the catalytic cycle rather than acetic acid[75]. Moreover, the construction of more reactive (acetato-κ$^2$O,O')(alkynyl)(phebox)rhodium(III) complexes from the original (diacetato)rhodium(III) complexes made the reaction reactivity limited (Fig. 4c)[76]. In 2017, the Ohshima group disclosed a catalytic strategy that enabled the rapid construction of *N*-unprotected α,α-disubstituted α-AA derivatives without the requirement of *N*-protective groups and avoiding the additional processes of protection and deprotection. This strategy was not only tolerable to diverse ketimines without the *N*-protection, but also to various carbonyl nucleophiles (Fig. 4d)[77]. Later, the same group has reported the other strategy for the synthesis of α,α-disubstituted α-AAs, involving the cyanation and hydrophosphonylation of in situ-formed *N*-unprotected ketimines[78]. Moreover, the Nakamura group disclosed that *N*-unprotected ketimines could also react with phosphine oxides to afford α,α-disubstituted α-AAs[79].

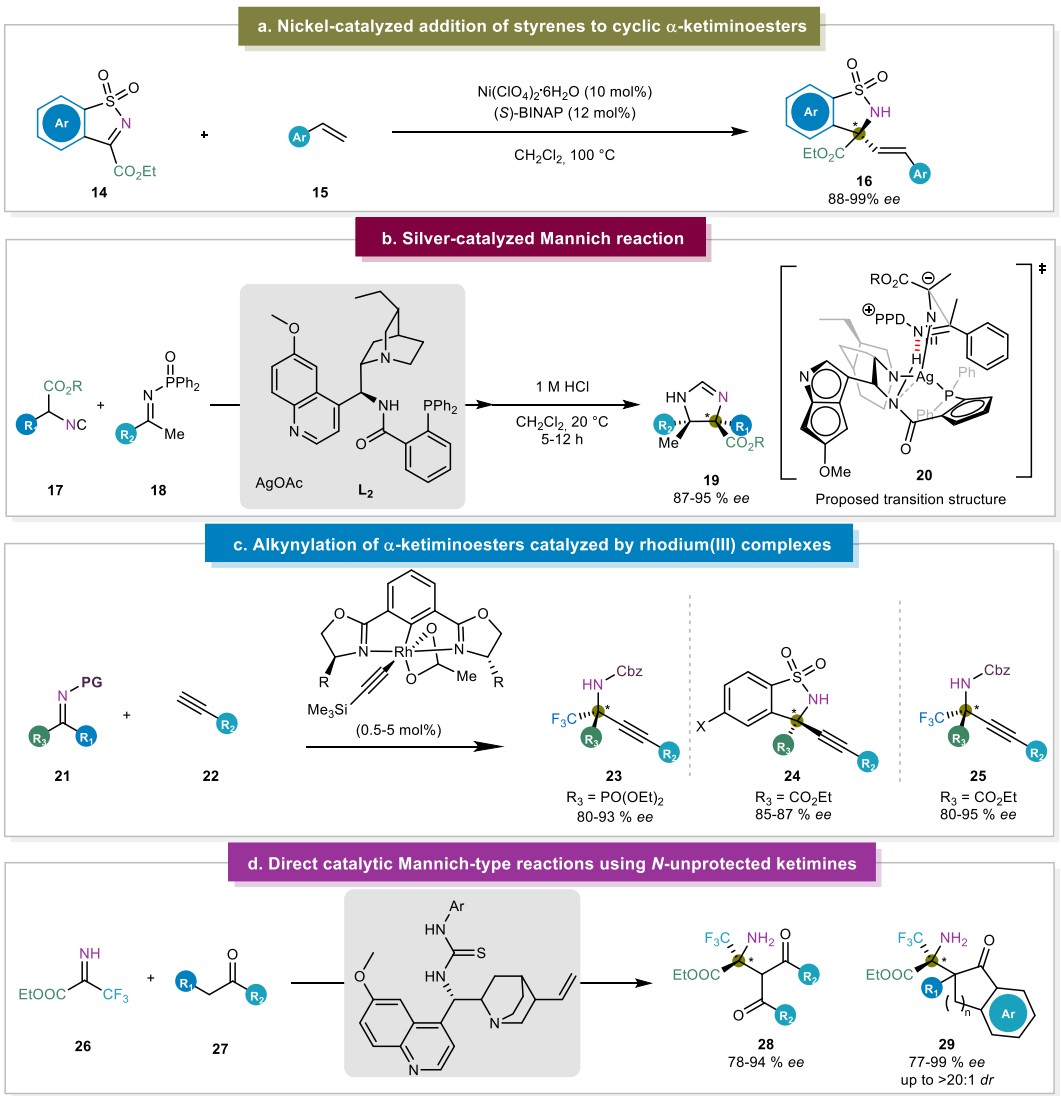

**Fig. 4 | Addition to ketiminoesters. a** Enantioselective addition of styrenes[25]. **b** Silver-catalyzed Mannich reaction of α-substituted isocyanoacetates and ketimines[26]. **c** Rhodium-catalyzed alkynylation of α-ketiminoesters[76]. **d** Direct access to *N*-unprotected α-tetrasubstituted amino acid esters[77].

A more recent approach from Procter and co-workers involved a borylative multi-component coupling using a Cu(I) catalyst (Fig. 5a)[28]. In the presence of a base, ketiminoesters **26**, terminal allenes **27** and B₂pin₂ **28** were converted into α,α-disubstituted α-amino esters **29** with moderate to high diastereomeric ratios. It should be noted that allenes have rarely been used in the construction of α,α-disubstituted α-AAs before this work. Moreover, when introducing 1,1-disubstituted allenes to the reaction, a change in regioselectivity was observed, which had not been previously reported in a catalytic allylcopper addition to imines. Additionally, a gram-scale reaction was performed, which provided 1.75 g of product in 83% yield and a *dr*-value of >95:5 after crystallization. However, some primary alkyl allenes bearing linear and branched alkyl groups furnished products in moderate diastereoselectivity. The proposed mechanism was believed to initiate with the generation of copper-alkoxide complex **a**, Afterwards, the transmetalation with B₂pin₂ yielded the borylcopper species **b**. Then the complex b underwent a complicated reaction process with allene **27** and resulted in the generation of allylcopper species as an intermediate. Because of the steric attributes of the NHC ligand, copper preferentially coordinates with the terminal double bond of the allene and performs addition to the less hindered terminal carbon, leading

to the formation of the *Z*-configured allylcopper **c**. The ensuing step involves the γ-addition of allylcopper **c** to the ketiminoester **26**, inducing the creation of two distinct stereocenters. The regeneration of the copper-alkoxide **a** or the borylcopper species **b** from the copperamide **d** ensues. Ultimately, the desired α,α-disubstituted α-amino esters are acquired by subjecting compound **29** to hydrolysis during the final workup phase.

A robust and modular approach to obtain all-alkyl α,α-disubstituted α-amino esters has been recently described by the group of Gaunt[29]. In this approach, a carbonyl alkylative amination reaction was initiated via blue light and a silane reagent which combined primary amines, α-ketoesters, and alkyl iodides in a single reaction. First, in situ condensation between the primary amine and α-ketoester was mediated by a Brønsted acid and formed the ketiminium species, which subsequently underwent rapid 1,2-addition of an alkyl radical to generate the corresponding aminium radical cation. The desired product was finally afforded from the electrophilic aminium radical cation, formed by a hydrogen atom transfer (HAT) process from electron-rich silane. This facile approach featured broad substrate scope with all the three components and generated impressive structurally diverse products (Fig. 5b). Another visible light approach was reported by the group of Gong.

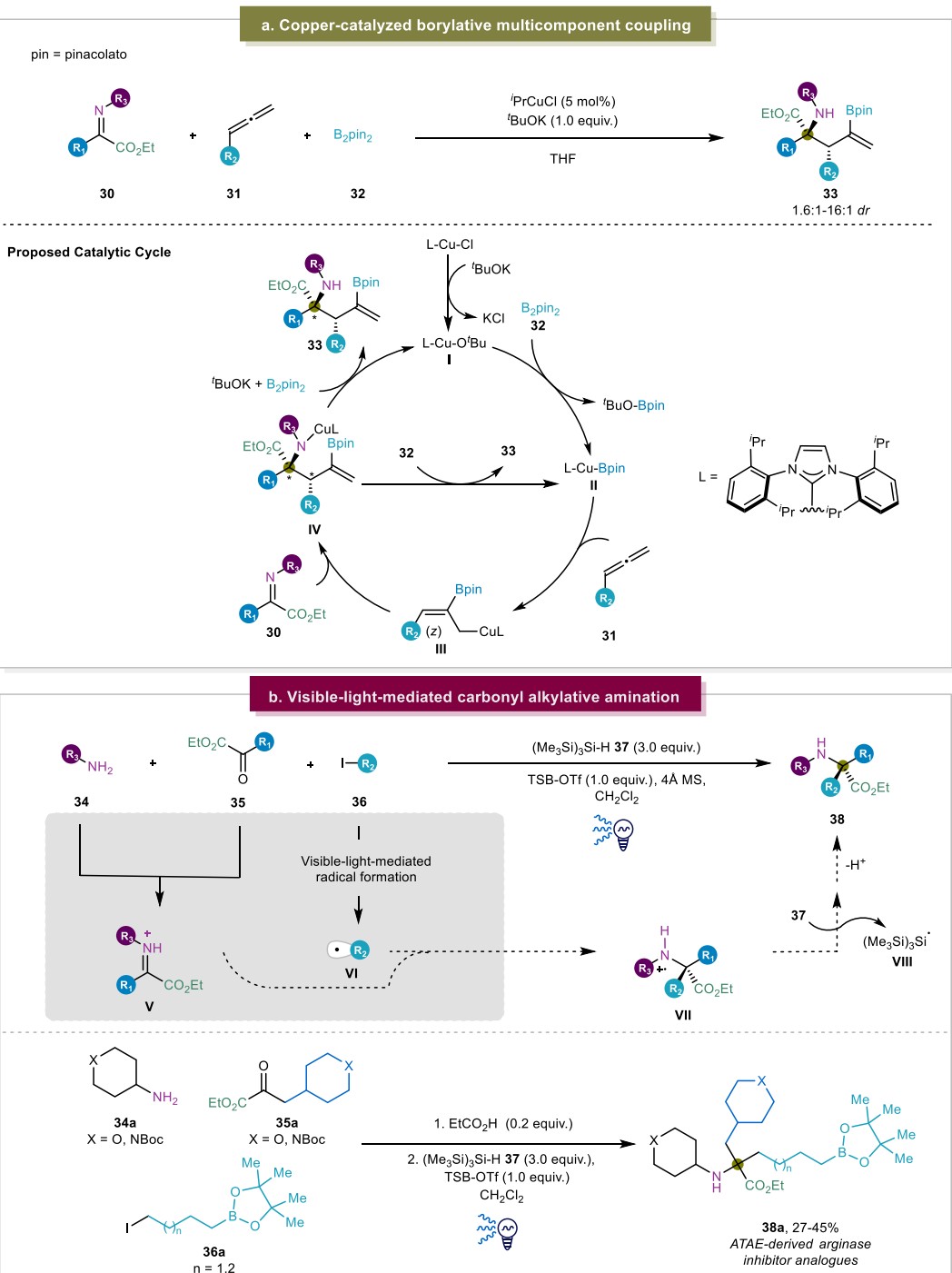

**Fig. 5 | Recent advances of addition at ketiminoesters. a** Procter's borylative multi-component coupling using a copper(I) catalyst[28]. **b** Visible-light-mediated three-component reaction to afford α,α-disubstituted α-amino esters[29].

In this case, the selective C(sp³)-H functionalization also proceeded through a HAT process. By adding a bisoxazoline-based (BOX) catalyst, the cross-coupling between the alkyl radical with imines enabled the desirable α,α-disubstituted α-AAs[80]. Very recently, alkyl radicals formed via an HAT step have also been reported to undergo radical addition to imines to for the α,α-disubstituted α-AAs[81]. Overall, these advances via addition at ketimines provided more structural diversity of afforded α,α-disubstituted α-AAs through reaction with previously challenging reagents such as styrenes and allenes. Moreover, during the preparation of our manuscript,

another extensive review about the catalytic enantioselective reactions of ketimine-type α-iminoesters/α-iminoamides to access unnatural α-AAs was published[82].

### Amino acid Schiff bases for α,α-disubstituted α-amino acid synthesis
α-Functionalization of benzophenone-based Schiff bases has been widely used for the construction of enantioenriched non-proteinogenic α-AAs. However, previous reports using Pd(0) as the catalyst together with achiral ligands failed to afford the α-AAs with

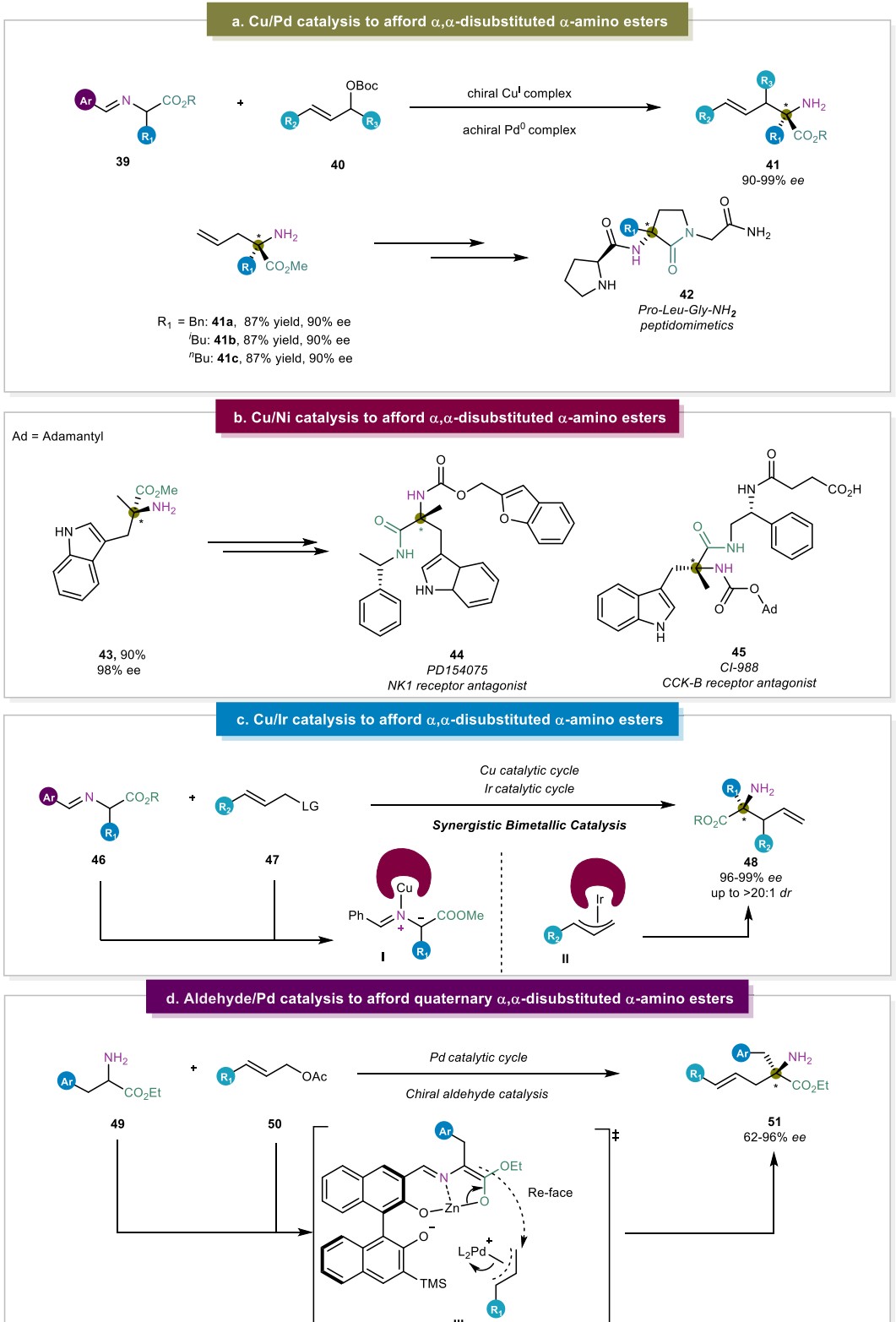

**Fig. 6 | Amino acid schiff bases for unnatural α-amino acid synthesis. a** Synergistic Cu/Pd catalysis[30]. **b** Synergistic Cu/Ni catalysis[31]. **c** Synergistic Cu/Ir catalysis[34]. **d** Aldehyde/Pd catalysis to afford α,α-disubstituted α-amino esters without the preparation of Schiff bases[35].

high enantioselectivities due to the distance between the stereogenic and catalytic centers[83–85]. To overcome these, Wang and co-workers disclosed a strategy which relied on the synergistic Cu/Pd catalysis to realize the enantioselective allylic alkylation of readily prepared aldimine esters (Fig. 6a)[30]. Their key to success was the interaction

between the achiral allylpalladium intermediate with the in-situ-generated α-substituted metallated azomethine ylide. The latter was activated by a chiral copper(I) complex, inducing the selective allylic alkylation with good enantioselectivity. More importantly, this strategy was further applied to the synthesis of a key intermediate of PLG

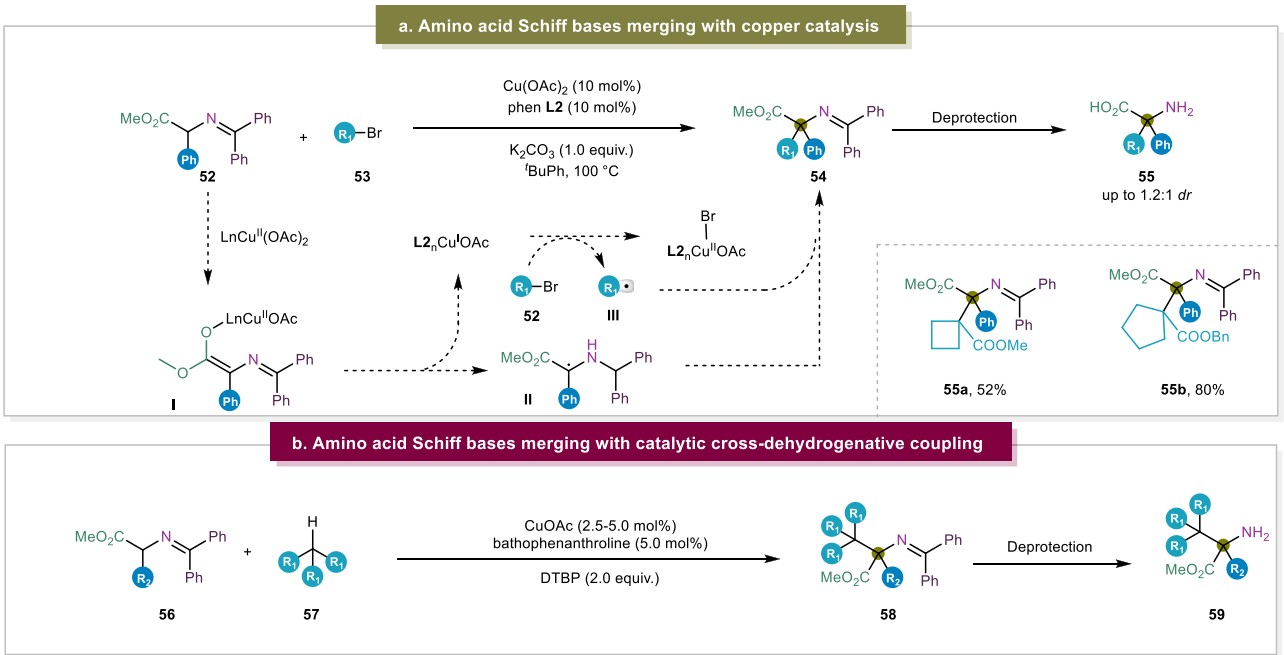

**Fig. 7 | Amino acid schiff bases for unnatural α-AA synthesis via radical coupling. a** Copper-catalyzed radical-radical coupling[13]. **b** Catalytic cross-dehydrogenative coupling[36].

(Pro-Leu-Gly-NH₂) peptidomimetics, a bioactive molecule with improved ability to modulate D2 dopamine receptors within the central nervous system. A similar strategy developed by the Zhang group, also utilized this Cu/Pd dual catalytic system[86]. One major difference in this procedure is that the palladium center carries a chiral ligand, as opposed to Wang's work, in which an achiral ligand was sufficient. However, Zhang's contribution, besides its broad substrate scope, additionally focused on the synthesis of di-, tri-, and even tetrapeptides with high stereoselectivity (up to >20:1 *dr* and 95% *ee*).

In 2022, the same group also reported the synergistic asymmetric Ni/Cu-catalyzed benzylation of aldimine esters[31]. With this robust method in hand, versatile benzyl-substituted α,α-disubstituted α-AAs were synthesized in high yields with satisfactory enantioselectivity. Afterwards, mechanistic studies were carried out, which suggested that the η³-benzylnickel intermediate with strong electrophilicity was essential to obtain high reactivity. Moreover, atom in molecule analysis of Ni/Cu and Pd/Cu complexes were performed and it was found that the former was more stable than the latter, suggesting that the Ni/Cu complex enabled the reaction under base-free conditions at room temperature. This strategy also enabled the facile construction of several drug candidates, including the key intermediate of the NK1 receptor antagonist PD154075 (**47**) and CI-988 (**48**), which is used as a CCK-B receptor antagonist (Fig. 6b).

The construction of α-AAs bearing vicinal stereocenters remained a challenge until the successful applications of synergistic bimetallic catalysis. One major accomplishment in this area was reported by the group of Zhang[31–33], after which the group of Wang[34] also made some remarkable contributions. Their methods offered the possibility of a single transformation to provide a range of α,α-disubstituted α-AAs with vicinal stereocenters and complete control over their relative and absolute configurations. As for the mechanism, Zhang proposed in 2018 that the copper complex activated the prochiral nucleophile to provide an *N*-metalated azomethine ylide which strongly enhanced the stereorecognition of the reaction (Fig. 6c). Moreover, the ylide intermediate was intercepted by the reactive allyliridium, allowing for stereochemical control of the chiral center.

More recently, the Guo group disclosed a methodology through the combination of a chiral BINOL aldehyde, ZnCl₂ as a Lewis acid, and a palladium complex to realize the synthesis of α,α-disubstituted α-amino esters without the *N*-protection of α-amino acid esters (Fig. 6d)[35]. They found that chiral BINOL aldehydes without hydroxy groups could not provide the desired reaction product. This strongly suggested that the existence of a hydroxy group at the 2′ position of the BINOL aldehyde was a key factor in this reaction, possibly owing to the coordination between the hydroxy group and π-allyl Pd(II). Indeed, utilization of synergistic strategies provided new avenues to reach to the *N*-protected and even *N*-unprotected α,α-disubstituted α-amino esters. Despite the remarkable advances in this reaction pattern, precious transition metals were required along with the synthesis of starting materials bearing activated or leaving groups.

In 2020, the Ohshima group developed a strategy that was different from the traditional synthesis using Schiff bases (Fig. 7a)[13]. In previous methods, Schiff bases generally only reacted with primary and very limited secondary alkyl halides mainly owing to the steric hindrance of tertiary alkyl groups. In this work, the amino acid products are the result of a radical-radical coupling of two radical species that are obtained through a one-electron process. Initially, via a base-mediated enolization of the substrate, the Cu(II)-catalyst is readily reduced to yield a Cu(I)-center and the radical Schiff base ester. Subsequently, the Cu(I)-species is oxidized back to the Cu(II) oxidation state as it reduces the alkyl halide. The resulting alkyl radical then undergoes a radical-radical coupling reaction with the open-shell Schiff base species to deliver the protected α,α-disubstituted α-amino ester in moderate to high yields. However, due to the radical nature of this mechanism, no stereoselectivity was observed. Therefore, the ester function of the substrate was modified with a chiral auxiliary which provided the protected α,α-disubstituted α-amino ester with high diastereoselectivity. Even though it is desired to employ tertiary alkyl bromides to react with Schiff base, the scope of tertiary alkyl bromides was quite limited. Very recently, Ohshima and coworkers continuously displayed an elegant protocol to obtain highly congested α-AAs and peptides with different tertiary hydrocarbons, this strategy

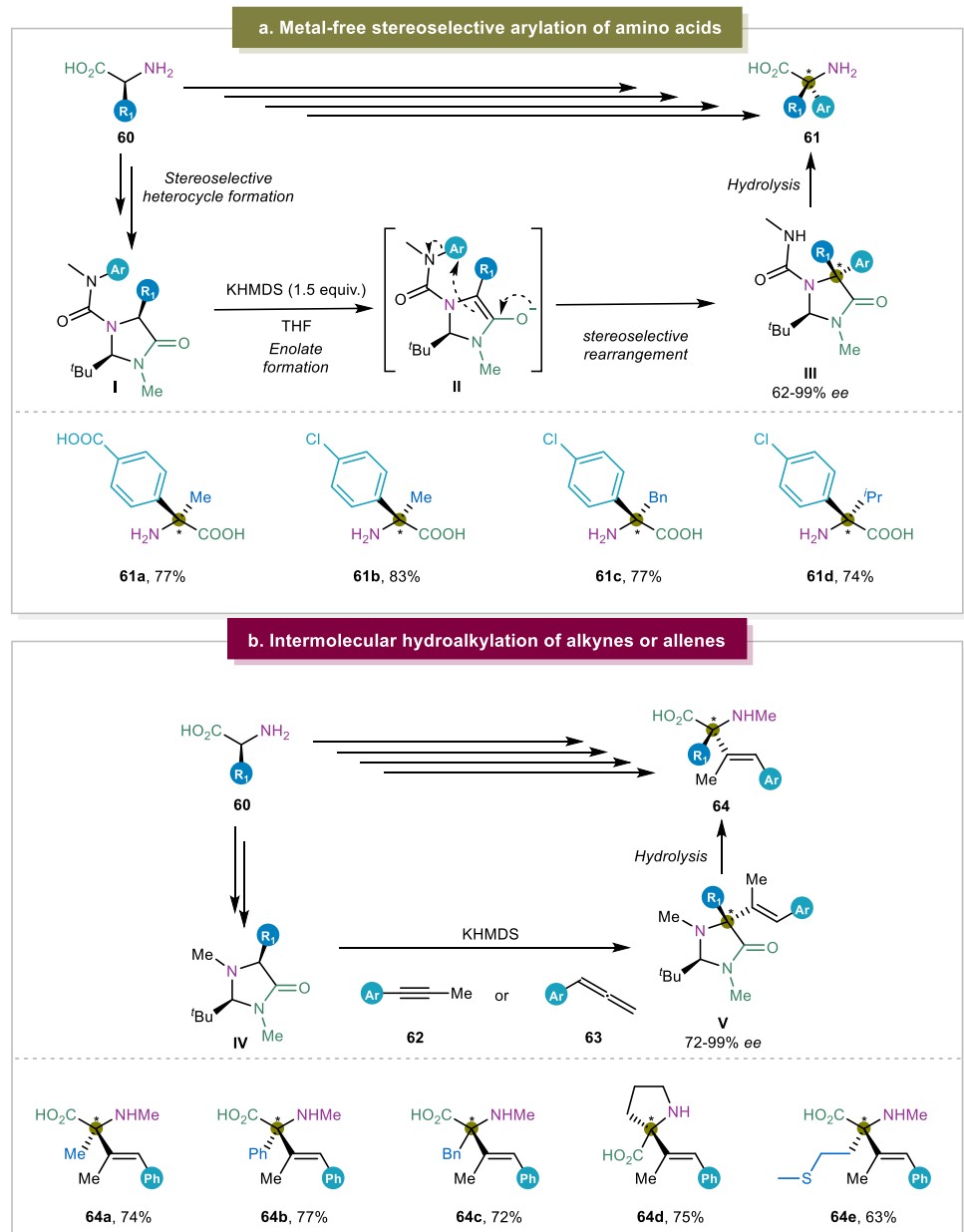

**Fig. 8 | α-Functionalization of general α-AAs. a** Metal-free arylation of amino acids[89]. **b** Hydroalkylation of alkynes and allenes[39].

obviated the use of typical alkyl halides to synthesize α,α-disubstituted α-AAs which is a tremendous advance (Fig. 7b)[36].

Overall, the modification of α-amino Schiff base methods provided various pathways featuring the incorporation of sterically demanding groups and introduction of contiguous α,α-disubstituted carbon centers. However, exploring synergistic catalysis using abundant metal catalysts or metal-free catalysts and strategies using unprotected substrates to improve the efficiency and atom economy are desirable to be reported in future. The copper-catalyzed radical-radical coupling also showcased the potential to use light irradiation as the energy source, which opens up the possibility for this field to move towards photochemical strategies, thereby replacing or even completely circumventing excessive oxidants.

## α-Functionalization of general α-AAs

In general, α-AAs undergo the functionalization at their α-position and this is an alternative effective way to access α,α-disubstituted α-AAs.

Even though the α-alkylation of available α-AAs is well-established, the arylation of available α-amino acids remains underexplored[87]. Recently, however, the group of Clayden reported the construction of α,α-disubstituted α-AAs via the α-arylation process (Fig. 8a)[88,89]. In this case, the N′-aryl ureas were firstly generated, serving as an intramolecular source of the coupling partner, and used for the following arylation reactions to achieve diverse N-carbamoylimidazolidinones. Afterwards, the key step of C-C bond formation was the rearrangement reaction to realize the arylation of the amino acid α-center in a diastereoselective manner. Moreover, this method featured practical applications and offered potential opportunity to synthesize α,α-disubstituted α-AAs in gram scale.

To further utilize the afforded α,α-disubstituted α-AAs in peptide synthesis especially incorporating this unit into the peptides via Fmoc solid-phase peptide synthesis (SPPS), a more practical and scalable approach to synthesize α,α-disubstituted α-AAs bearing aryl groups was introduced by the same group. The stability and tolerance of α,α-

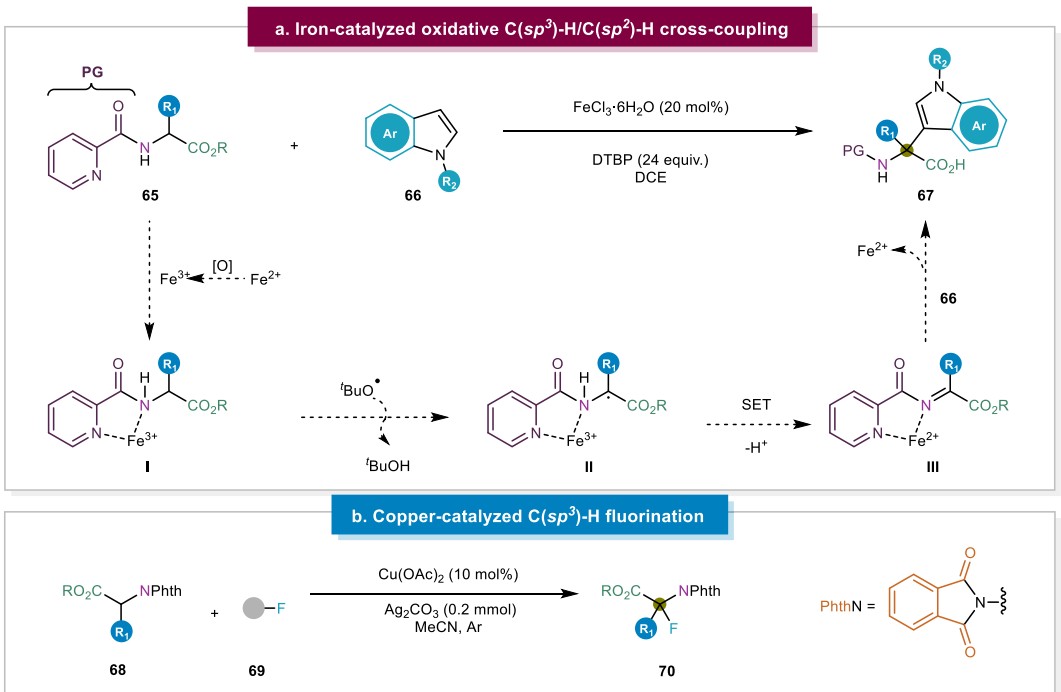

**Fig. 9 | Direct α-C($sp^3$)-H bond functionalization. a** Iron-catalyzed C($sp^3$)-H/C($sp^2$)-H cross-coupling[94]. **b** Copper-catalyzed C($sp^3$)-H direct fluorination[95].

disubstituted α-AAs in a α-helical peptide was verified, which showcased the potential utility of such α,α-disubstituted α-AAs via arylation of other proteinogenic α-AAs[37]. Beside the arylation of the general amino acids, the same group also disclosed the α-alkenylation of general α-amino acids using similar strategy to extend the applications of this approach[38]. Later, α-alkenylation of general α-amino acids was also reported by the Breit group to furnish the α,α-disubstituted α-alkenyl substituted amino acids[39]. The difference compared to the previous report from the group of Clayden was the achievement of intermolecular α-alkenylation (Fig. 8b).

In conclusion, the functionalization of general α-amino acids has recently achieved great progress to construct the unnatural α,α-disubstituted α-AAs. However, this strategy requires the additional preparation of *N*-alkenyl urea substrate or imidazolidinone, which reduces the step economy. Moreover, strong metal base is required. Therefore, one-pot or less steps to achieve this reaction is anticipated.

## Direct α-C-H bond functionalization of α-amino acids to afford α,α-disubstituted α-AAs

Over the last decades, C($sp^3$)-H bond functionalization via transition-metal catalysis has been recognized as a robust synthetic strategy to construct C-C bonds[90–93]. The key to achieving such an ideal transformation is to activate inert C($sp^3$)-H bonds and to investigate the right coupling partners. In this respect, direct C($sp^3$)-H bond functionalization via coordination activation strategy is an emerging strategy, especially through the metal-catalyzed oxidative functionalization of α-C($sp^3$)-H bonds of α-tertiary α-amino acid/esters.

In 2013, You and co-workers disclosed their strategy to synthesize β-aromatic α,α-disubstituted α-amino esters via a Ni-catalyzed C($sp^3$)-H/C($sp^3$)-H cross-coupling reaction[94]. The presence of a 2-pyridinecarbonyl moiety on the substrate, together with an amine nitrogen atom, facilitated the coordination of the nickel center to allow for the activation of the targeted α-C-H bond. This protocol involving a diradical intermediate tolerated a broad range of substrates to assemble α,α-disubstituted α-amino acids. Inspired by this work, functionalization of the α-position of tertiary α-amino esters was further described, wherein the authors focused on an iron-catalyzed

C($sp^3$)-H/C($sp^2$)-H cross-coupling (Fig. 9a)[40]. A similar reaction mechanism was proposed in their previously mentioned work about the C($sp^3$)-H/C($sp^3$)-H cross-coupling[94], wherein a 2-pyridinecarbonyl group facilitates coordination with the active Ni(III) metal center. Subsequently, a *tert*-butoxy radical abstracts the α-hydrogen atom, delivering a stabilized open-shell intermediate. This species initiates an intramolecular SET to finally release Fe(II) and the product upon attack by the coupling partner on the electrophilic α-position of the coordination complex. Instead of introducing carbon-based substituents, the Liu group developed a synthetic route for α-fluorinated α-AA derivatives via a copper-catalyzed C($sp^3$)-H direct fluorination (Fig. 9b)[95]. This strategy provided broad substrates scopes with moderate to excellent yields including aromatic, as well as aliphatic substrates. They also provided an efficient removal of the amide auxiliary group, unlike previous reports where this was impossible without affecting the newly formed C-F bond[96,97]. Although a Cu(II)-catalyzed SET oxidative addition mechanism was revealed, asymmetric fluorination of the α-AA derivatives has still not been investigated.

In summary, the direct α-C($sp^3$)-H bond functionalization remains a valuable tool for the synthesis of α,α-disubstituted α-AA's. However, there are still some limitations, such as requirement of expensive transition metal catalysts, excess oxidants, as well as the lack of stereoselectivity. Although the coordinating group can be removed easily, additional reaction steps are needed.

## Hydrocarboxylation of amines or imines to form the α,α-disubstituted α-amino acid derivatives

In the last decade, visible-light-mediated transformations have emerged as a new and green paradigm in organic synthesis[98–101]. With this robust and relatively benign alternative, numerous reactions can proceed which were usually thermally unachievable. The advancement of photochemical transformations, including photoredox catalysis, energy transfer, and hydrogen atom transfer, has been employed in the synthesis of α,α-disubstituted α-AAs successfully[41,102]. Nowadays $CO_2$ has been used as a C1 building block due to its nontoxicity and abundant availability[103–108]. In particular, $CO_2$ serves as a renewable carbonyl source, which was used by the group of Yu in the catalytic

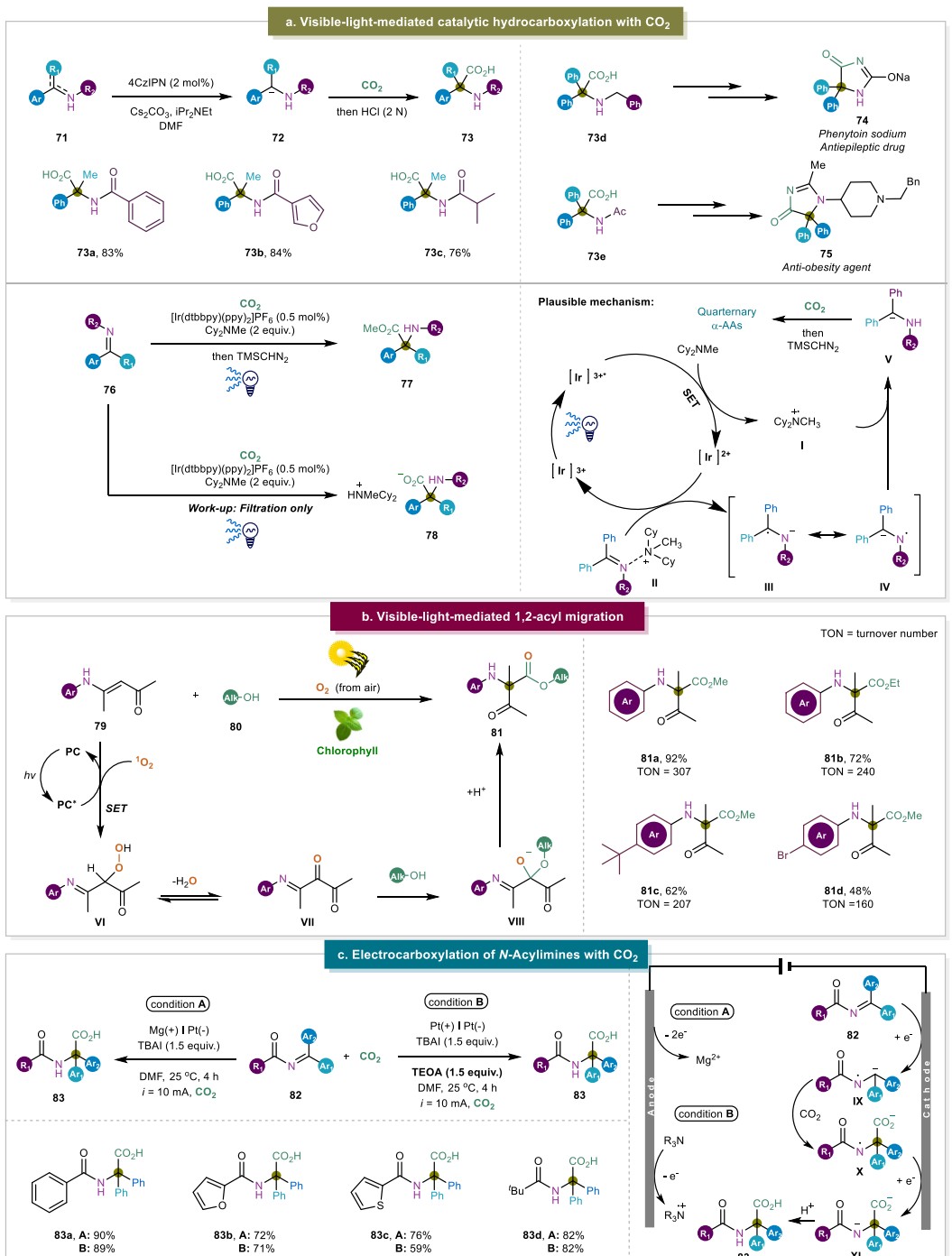

**Fig. 10 | Hydrocarboxylation of amines or imines. a** Visible-light-mediated $CO_2$-fixation; PC: photocatalyst (4-CzIPN)[41]. **b** Visible-light promoted 1,2-acyl migration[102]. **c** Electrocarboxylation of N-acylimines[112].

hydrocarboxylation of enamides or imines to afford α,α-disubstituted α-AAs (Fig. 10a)[41]. This was achieved via the light-mediated reduction of enamides to form the corresponding α-amino carbanions as intermediates. This was followed by trapping $CO_2$ and protonation to provide the desired α,α-disubstituted α-AAs. This reaction showcased the potential of $CO_2$ fixation in the synthesis of α,α-disubstituted α-AAs. However, this strategy faced a few challenges in the design of imines and enamides to reach a feasible reductive potential, as this reduction is key in order to generate the carbanion that captures $CO_2$. Around that time, the group of Walsh reported the visible-light-

promoted $CO_2$-fixation with imines to synthesize diaryl α,α-disubstituted α-AAs which featured a similar reaction pathway[109].

Walsh's work provided detailed mechanistic studies by combining UV-Vis spectra, Stern–Volmer fluorescence quenching experiments. A plausible mechanism is also depicted where at first, the photocatalyst absorbs visible light to reach to the excited state. Then the tertiary amine acts as an electron donor and reduced $[Ir]^{3+*}$ to $[Ir]^{2+}$ to form the radical cation **I**. The $[Ir]^{2+}$ further reduces the imines to form the radical anion **III/IV**, which is quenched by the radical cation **I** to generate the carbanion. The formed carbanion reacts with $CO_2$ to furnish the final

product. Additionally, the group of Yu displayed the phosphono-carboxylation of alkenes using $CO_2$ to construct the α,α-disubstituted α-AAs[110]. Compared to previous reports, this work brought more opportunities to construct unusual α,α-disubstituted α-AAs via the difunctionalization of enamides or alkenes.

Light-mediated 1,2-acyl migration was also successfully applied into the field of α,α-disubstituted α-AAs synthesis. The Li group firstly disclosed that secondary enaminones could be oxidized by light-induced singlet oxygen. After the subsequent nucleophilic addition of alcohols, the desired α,α-disubstituted α-amino esters were generated via the following 1,2-acyl migration[111]. Parallel to this work, our group found that chlorophyll, a sustainable and cheap photocatalyst, can be employed in the generation of singlet oxygen efficiently, and subsequently underwent the 1,2-acyl migration to afford the desired α-amino esters (Fig. 10b)[102]. More importantly, this reaction was also driven via solar energy in good yields. However, there are still several challenges and difficulties that need to be addressed in the future such as poor diversity of enaminones and less reactivity with the growing carbon number of employed alcohols.

Recently, the Lu group has displayed an elegant strategy enabling the electrocarboxylation of *N*-acylimines with $CO_2$ to afford valuable α,α-disubstituted α-AAs. Notably, this electrochemical method could be carried out without sacrificial anodes and triethanolamine served as an external reductant (Fig. 10c)[112]. Besides of these approaches, it is anticipated that the expansion of the difunctionalization of alkenes will enrich the structural diversity of α,α-disubstituted α-AAs via using other radical precursors[113,114].

## Outlook and conclusions

This review has demonstrated that the recent emerging strategies have become attractive and valid alternatives to afford α,α-disubstituted α-amino acid derivatives. Moreover, some strategies such as direct $CO_2$ fixation to construct a carboxylic group, construction of highly congested α,α-disubstituted α-AAs and α,α-disubstituted α-AAs bearing vicinal stereocenters are achieved using the emerging toolbox which were difficult using the typical synthetic methods. With such robust methods, structural diversity and efficacy was achieved and subsequently boosted the practical applications of α,α-disubstituted α-AAs specifically in drug discovery. It is also desirable that new strategies are not only helpful on the development of efficient reaction processes, but also make great contributions to introduce α,α-disubstituted α-AAs with higher atom economy and sustainability.

With these strategies in hand, diverse α,α-disubstituted α-AAs could be obtained. However, the incorporation of α,α-disubstituted α-AAs into peptides would expand the utility and value of these new methods. Herein, the application of α,α-disubstituted α-AAs has been briefly discussed to address the outlook. It is established that unnatural α,α-disubstituted α-AAs can be introduced into peptides successfully as they have been used as suitable substrates to undergo the *N*-terminal functionalization via chiral phase-transfer catalysis[115]. However, this worked only showed the feasibility for the dipeptide synthesis. Later, more complicated peptides bearing α,α-disubstituted α-AAs have been reported which showed potential bioactivities[116–118]. Moreover, α,α-disubstituted α-AAs have been successfully applied into the peptide synthesis via solid-phase synthesis[119]. It should be also noted that the reactivity of α,α-disubstituted α-amino acids in direct peptide coupling reactions is typically low due to the steric hindrance. Therefore, the development of new peptide coupling reactions using α,α-disubstituted α-amino acids as substrates is an important topic in future. It is also promising to merge the state-of-the-art with typical methods to overcome some limitations which are encountered in typical methods. For example, the Ohshima group has developed several brilliant strategies via the combination of amino acids Schiff bases with radical methods[120,121]. Therefore, more efforts and outcomes are anticipated for the design of α,α-disubstituted α-AAs based on the emerging toolbox. In addition, most of established methods require the protection of amino groups or carboxylic, even both of them. Therefore, strategies without the protection of functional groups are more practical and valuable[122]. Moreover, the installation of α,α-disubstituted α-AAs into peptides is also fascinating to realize the functionalization of peptides and even proteins[123]. It should be noted that electrochemistry, including photoelectrocatalysis, has been also prevalent recently as a robust tool[124–126]. Although many challenging organic transformations are achieved via electrochemistry[127], using this alternative to construct α,α-disubstituted amino acids are rarely explored[128].

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

## Acknowledgements

We thank the Francqui foundation (to S.D.), the Chenguang Program of Shanghai Education Development Foundation and Shanghai Municipal Education Commission (22CGA51, to Y.Z.), Dehausse fellowship (to J.V.) for their generous support and FWO for the Odysseus grant (to S.D.).

## Author contributions

S.D. conceived the project; Y. Z., J. W. and J. V. wrote the manuscript; S.D. guided and reviewed the manuscript; Y. Z. and J. V. contributed equally to the article.

## Competing interests

The authors declare no competing interests.
