## [Peer Review File · Nature Communications]

Reviewers' comments:

Reviewer #1 (Remarks to the Author):

This review paper summarizing the latest synthesis methods for α,α -disubstituted α -amino acids appears to be a valuable contribution to the field. The paper comprehensively covers major reactions since 2015, providing insight into the state-of-the-art in α,α -disubstituted α -amino acid synthesis. Given the growing interest in unnaturalized peptides as a new modality in drug development, this review will greatly interest researchers in this field.

However, the following points need improvement and clarification:

Overall Comments:

Terminology: Is the term "Quaternary α -Amino Acids" appropriate? When referring to carbon centers, all carbons with four substituents are quaternary carbons. However, when heteroatoms are involved, it is more accurate to refer to them as "tetrasubstituted carbons." Please verify and correct this terminology.

Figure Explanations: Some of the figure descriptions lack clarity, making it difficult to understand the connections between the figures and their corresponding text explanations. It is essential to provide clear references and descriptions for each figure to enhance the readers' understanding.

Color Usage: Be mindful of using colors to distinguish information, as some readers might have color perception issues. Consider using other methods, such as clear labeling or patterns, to convey information without relying solely on colors. For example, in Figure 8a, it is difficult to recognize which substitution group, Ar or alkyl, originates from the amino acid precursor. It would be beneficial to clarify this by using a similar approach as shown in Figure 8b, where one of the substitution groups is explicitly labeled as "Ar."

The extensive use of colors, especially shades of blue and green, as well as red and purple, makes it difficult to differentiate between certain elements in the figures. Please consider using more distinguishable colors or patterns to enhance clarity, taking into account that some readers may have difficulty discerning between specific color variations. In addition, it is particularly difficult to distinguish between colors such as blue and green or red and purple.

Compound Numbering: There are instances of incorrect usage of parentheses with compound numbers throughout the manuscript. Parentheses should be reserved for indicating compound numbers following the formal names, as demonstrated in the example PD154075 (47). Therefore, the usage of parentheses, such as "carbonyl product (3)," is incorrect, and parentheses are not necessary in this context. Please revise the compound numbering consistently to adhere to the correct format.

The application of the synthesized compounds in pharmaceutical synthesis is briefly discussed. It would be beneficial to include more examples to showcase the utility of the developed reactions.

Specific Comments:

Introduction:

The difficulty in reacting with ketimines is attributed to steric hindrance, but it's important to note that electronic factors also play a significant role. Ketimines are known to exhibit lower electrophilicity compared to aldimines.

Figure 1a:

The colors of the three circles in the upper-left corner do not correspond to the amino acid structures above, making it unclear what they represent. Additionally, the term " α " is erroneously written as "a."

Figure 1b:

This figure should explain the general two-electron-type ion reactions (e.g., phase-transfer catalysis). In that case, the imine moiety should be an aldimine. Also, the necessity of using aldimines for ion reactions and ketimines for radical reactions should be explained somewhere in the paper.

The hydrolysis step is missing in the transformation from the Strecker reaction product to the amino acid.

Figure 2:

The statement "the electrophilicity of the α -carbon is increased through coordination with transition metals" lacks clarity when looking at the figure.

The figures classify C-/N+ and C+/N- reactions, but it is essential to consider that C-N bond formation often involves reactions with radical species, as illustrated in Figure 3a.

Additionally, the correct term should be "addition to ketimines" instead of "addition of ketimines."

Figure 4:

Catalytic Asymmetric Nucleophilic Addition to Ketimines:

The catalytic asymmetric nucleophilic addition reaction using N-unprotected ketimines as substrates has gained significant attention recently, with various reactions reported by Ohshima, Nakamura, and others. Considering the importance of these reactions, it would be beneficial to include relevant descriptions and figures in the paper.

CO₂ Utilization Reactions:

Reactions utilizing CO₂ include not only photochemical reactions but also electrochemical reactions, which have been reported. It would be valuable to provide explanations and examples of both types of reactions.

Importance of α,α -Di-Substituted α -Amino Acids in Peptide Synthesis:

From the perspective of peptide synthesis involving α,α -di-substituted α -amino acids, the following three aspects are crucial:

Can they be converted into Fmoc amino acids for solid-phase synthesis?

Can peptides (at least dipeptides) be used as substrates?

Can they be successfully incorporated into peptide chains via peptide coupling? It is worth noting that due to steric hindrance, the reactivity of α,α -di-substituted α -amino acids in peptide coupling reactions is typically low. The development of new peptide coupling reactions using α,α -di-substituted α -amino acids as substrates is an important research topic for the future.

These comments should help improve the overall quality and clarity of the manuscript. Please address each point to enhance the value of the review paper on the synthesis of α,α -di-substituted α -amino acids.

Reviewer #2 (Remarks to the Author):

Amino acid is fundamental component of life. The motif of amino acid appears in many bioactive molecules such as native products and drugs. In addition, since the nature chiral amino acid is readily available, it is widely utilized as basic chiral unit in modifying molecules such as chiral ligand and organic functional materials. Given its high value in organic synthesis, accessing to diversity of amino acids especially Quaternary α -Amino Acids is of great importance and has been attracting a large number of scientists. Thus, it is worthwhile providing an overview of this field. In the review, the author summarized the recent advances in synthesizing Quaternary α -Amino Acids. Different reaction types are introduced including synergistic catalysis, C-H activation, CO₂ fixation, and photocatalysis. The development of each type is comprehensively introduced and the content is basically completed. Therefore, we believe that this manuscript can be published, but the following questions are still needed to be addressed:

1. Some key mechanisms are supposed to be clearly described in order to make this manuscript more accessible. (. Figure 3a, 3d-r; Figure 4-b; Figure 5a; e.g. Figure 3a, the “radical addition” step is confusing: what is the source of the new carbonyl group? How does it be connected to III?).
2. The yields and stereoselectivities (ee or dr) of all the corresponding products (Figures 1-10) should be added to give the readers more useful information.
3. Page 11, “However, previous reports using Pd(0) as the catalyst together with achiral ligands suffered from affording the α -AAs with high enantioselectivities.”. This sentence is confusing. Why do high enantioselectivities make it suffer?
4. One important work about synthesis of chiral α -AAs from amino acid Schiff bases should be cited: J. Am. Chem. Soc. 2017, 139, 29, 9819–9822.
5. Figure 10a, the equiv. of Cy₂NMe is missed.
6. Ref. 62. The abbreviation of journal name is wrong.

Responses to Reviewer 1

Comment 1: Terminology: Is the term "quaternary α -Amino Acids" appropriate? When referring to carbon centers, all carbons with four substituents are quaternary carbons. However, when heteroatoms are involved, it is more accurate to refer to them as "tetrasubstituted carbons." Please verify and correct this terminology.

Response: We appreciate the referee for this comment. We have changed the 'quaternary α -amino acid' to ' α,α -disubstituted α -amino acids' in the entire manuscript.

Comment 2: Figure Explanations: Some of the figure descriptions lack clarity, making it difficult to understand the connections between the figures and their corresponding text explanations. It is essential to provide clear references and descriptions for each figure to enhance the readers' understanding.

Response: We appreciate the referee for giving this valuable point. We have to make the content concise in the beginning to meet the words limitation of the journal. But it is true that makes some description unclear. We have already added the clear references for each figure and added more details in each figure. All the added description is marked with yellow color in the manuscript.

Comment 3: Color Usage: Be mindful of using colors to distinguish information, as some readers might have color perception issues. Consider using other methods, such as clear labeling or patterns, to convey information without relying solely on colors.

For example, in Figure 8a, it is difficult to recognize which substitution group, Ar or alkyl, originates from the amino acid precursor. It would be beneficial to clarify this by using a similar approach as shown in Figure 8b, where one of the substitution groups is explicitly labeled as "Ar." The extensive use of colors, especially shades of blue and green, as well as red and purple, makes it difficult to differentiate between certain elements in the figures. Please consider using more distinguishable colors or patterns to enhance clarity, taking into account that some readers may have difficulty discerning between specific color variations. In addition, it is particularly difficult to distinguish between colors such as blue and green or red and purple.

Response: We agree with the referee and appreciate his comment. We have made sure that there is no information left that solely depends on color differences.

Comment 4: Compound Numbering: There are instances of incorrect usage of parentheses with compound numbers throughout the manuscript. Parentheses should be reserved for indicating compound numbers following the formal names, as demonstrated in the example PD154075 (47). Therefore, the usage of parentheses, such as "carbonyl product (3)," is incorrect, and parentheses are not necessary in this context. Please revise the compound numbering consistently to adhere to the correct format. The application of the synthesized compounds in pharmaceutical synthesis is briefly discussed. It would be beneficial to include more examples to showcase the utility of the developed reactions.

Response: We appreciate the referee for this comment. All the numbering issues have been resolved in the updated version. We also agree that pharmaceutical applications increase the attractiveness of this review. For this reason, most of the presented reactions that contained pharmaceutical applications in the original report, contain these applications in the respective figure.

Comment 6: Introduction: The difficulty in reacting with ketimines is attributed to steric hindrance, but it's important to note that electronic factors also play a significant role. Ketimines are known to exhibit lower electrophilicity compared to aldimines.

Response: We appreciate this comment and we have added the following clarification into the manuscript: *The challenge in employing the Strecker reaction for the synthesis of α,α -disubstituted α -AAs also stems from the lower electrophilicity of the iminyl carbon of ketimines compared to aldimines*

Comment 7: Figure 1a: The colors of the three circles in the upper-left corner do not correspond to the amino acid structures above, making it unclear what they represent. Additionally, the term " α " is erroneously written as "a."

Response: We appreciate this comment and have verified the consistent color and pattern scheme. The ' α ' symbol is also corrected.

Comment 8: Figure 1b: This figure should explain the general two-electron-type ion reactions (e.g., phase-transfer catalysis). In that case, the imine moiety should be an aldimine. Also, the necessity of using aldimines for ion reactions and ketimines for radical reactions should be explained somewhere in the paper.

The hydrolysis step is missing in the transformation from the Strecker reaction product to the amino acid.

Response: Thank you for providing this highly valuable suggestion. We have added the two-electron-type ion reactions into the figure 1b, the imine moiety as aldimine has been added. And we also explained the two-electron-type ion reactions including the phase-transfer catalysis in the introduction marked with yellow color. Regarding the ketimines with radical reactions, these reactions have been displayed in the Figure 7 with details.

The addition in the manuscript was shown below:

Afterwards, the coupling of enolates of Schiff-base-derived α -amino esters with diverse electrophiles has been reported for the formation of α,α -disubstituted α -amino acids.¹¹⁻¹² The difference in electrophilicity of the iminyl carbon also indirectly influences these types of reactions, as the reaction site, either anionic or radical in nature, is in direct conjugation with the imine. Therefore, these intermediates will have improved stabilization in aldimines and ketimines, respectively, due to the amount of electron-rich phenyl substituents.³⁵ The foundation of phase-transfer catalysis with achiral Schiff base esters, however, was laid by O'Donnell and Ghosez to afford dialkylated α -amino acids in 1982.¹³⁻¹⁵ This method allowed mild conditions and the ability to perform reactions on a large scale, however, the prior assembly of Schiff base esters was required.

In 2020, the Ohshima group developed a strategy that was different from the traditional synthesis using Schiff bases (Figure 7a). In previous methods, using Schiff base has been verified that only reacted with primary and very limited secondary alkyl halides mainly owing to the steric hinderance of tertiary alkyl groups. In this work, the amino acid products were the result of a radical-radical coupling of two radical species that were obtained through a one-electron process. Initially, via a base-mediated enolization of the substrate, the Cu(II)-catalyst was readily reduced to yield a Cu(I)-center and the radical Schiff base ester. Subsequently, the Cu(I)-species was oxidized back to the Cu(II) oxidation state as it reduced the alkyl halide. The resulting alkyl radical then underwent a radical-radical coupling reaction with the open-shell Schiff base species to deliver the protected α,α -disubstituted α -amino ester in moderate to high yields.

Comment 9: Figure 2: The statement "the electrophilicity of the α -carbon is increased through coordination with transition metals" lacks clarity when looking at the figure. The figures classify C-/N+ and C+/N- reactions, but it is essential to consider that C-N bond formation often involves reactions with radical species, as illustrated in Figure 3a. Additionally, the correct term should be "addition to ketimines" instead of "addition of ketimines."

Response: We appreciate this comment and have added some clarification by using arrows. This way, it is clear that the nucleophilic nitrogen substrate is able to attack a position which is otherwise unfeasible due to insufficient electrophilicity. The authors also agree that the radical pathway should indeed be mentioned. Therefore, the authors added some nuance and deleted the C⁻ and C⁺ labels in the figure as this does indeed not correctly represent the reaction types. The labels N⁻ and N⁺ have been retained since we still differentiate between N-electrophiles and -nucleophiles. Moreover, the "addition of ketimines." has been corrected.

Comment 10: Figure 4: Catalytic Asymmetric Nucleophilic Addition to Ketimines: The catalytic asymmetric nucleophilic addition reaction using *N*-unprotected ketimines as substrates has gained significant attention recently, with various reactions reported by Ohshima, Nakamura, and others. Considering the importance of these reactions, it would be beneficial to include relevant descriptions and figures in the paper.

Response: We appreciate the referee for giving this valuable point. Indeed, these strategies are highly important, and we have already added into our manuscript as shown in Figure 4d.

The addition in the manuscript is shown below:

*In 2017, the Ohshima group developed new catalytic reactions that enable the direct synthesis of *N*-unprotected α -tetrasubstituted amino acid esters under proton-transfer conditions, eliminating the need for *N*-protective groups and avoiding protection/deprotection steps. This strategy is applicable to a wide range of *N*-unprotected trifluoromethyl ketimines and carbonyl nucleophiles, including 1,3-diketones, malonates, α -ketonitriles, α -ketoesters, and 3-substituted oxindoles (Figure 4d).⁷⁷ Furthermore, the same group have developed a one-pot catalytic synthesis method for α,α -disubstituted α -AA derivatives. This method involves the cyanation or*

hydrophosphonylation of in situ-generated *N*-unsubstituted ketimines.⁷⁸ Moreover, the Nakamura group disclosed the reaction of *N*-unprotected ketimines with phosphine oxides to afford α,α -disubstituted α -AA derivatives.⁷⁹

Comment 11: CO₂ Utilization Reactions: Reactions utilizing CO₂ include not only photochemical reactions but also electrochemical reactions, which have been reported. It would be valuable to provide explanations and examples of both types of reactions.

Response: We appreciate the referee for this comment. We have added this CO₂ utilization reaction using electrochemical methods (*Org. Lett.* **2022**, *24*, 3565-3569) into the manuscript including the explanation and figure (**Figure 10c**).

The addition in the manuscript is shown below:

*Recently, Lu and co-workers have successfully developed an efficient electrocarboxylation of easily available *N*-acylimines with atmospheric CO₂ to prepare α , α -disubstituted α -amino acids. This work provided detailed mechanistic studies by combining cyclic voltammetry, and isotope labeling experiments. This reaction can be carried out without the need for sacrificial anodes, with triethanolamine serving as an external reductant (**Figure 10c**).*

Comment 12: Importance of α,α -di-Substituted α -Amino Acids in Peptide Synthesis: From the perspective of peptide synthesis involving α,α -di-substituted α -amino acids, the following three aspects are crucial: Can they be converted into Fmoc amino acids for solid-phase synthesis? Can peptides (at least dipeptides) be used as substrates? Can they be successfully incorporated into peptide chains via peptide coupling? It is worth noting that due to steric hindrance, the reactivity of α,α -di-substituted α -amino acids in peptide coupling reactions is typically low. The development of new peptide coupling reactions using α,α -di-substituted α -amino acids as substrates is an important research topic for the future.

Response: We appreciate the referee for giving this valuable suggestion. We have added the perspective of peptide synthesis involving α,α -disubstituted α -amino acids into our manuscript as shown below:

*With these novel strategies in hand, diverse α,α -disubstituted α -AAs with unprecedented substituents are obtained. However, the incorporation of α,α -disubstituted α -AAs into peptides With these novel strategies in hand, diverse α,α -disubstituted α -AAs with unprecedented substituents are obtained. However, the incorporation of α,α -disubstituted α -AAs into peptides would expand the utility and value of these new methods. Herein, the application of α,α -disubstituted α -AAs has been briefly discussed to address the outlook. First, it was known that unnatural α,α -disubstituted α -AAs have been introduced into the peptides successfully. First, the peptides have been proved as suitable substrates to undergo the *N*-terminal functionalization via the chiral phase-transfer catalysis and finally afforded the peptides bearing α,α -disubstituted α -AAs. But this worked only showed the feasibility for the dipeptides. Later, more complicated peptides bearing α,α -disubstituted α -AAs have been reported which showed potential bioactivities. Moreover, α,α -disubstituted α -AAs have been successfully applied into the peptide synthesis via solid-phase synthesis. The Hammer group has been reported that cyclic α,α -disubstituted α -AAs could be converted to the Fmoc-amino acids and further applied to the automated solid-phase Fmoc chemistry. Therefore, we believe the synthetic development of α,α -disubstituted α -AAs would benefit the peptide synthesis and related drug discovery. It should be also noted that the reactivity of α,α -disubstituted α -amino acids in direct peptide coupling reactions is typically low due to the steric hindrance. Therefore, the development of new peptide coupling reactions using α,α -disubstituted α -amino acids as substrates is an important topic in future.*

Comments from reviewer 2 and our reply:

Comment 1. Some key mechanisms are supposed to be clearly described in order to make this manuscript more accessible. (. Figure 3a, 3d-r; Figure 4-b; Figure 5a; e.g. Figure 3a, the “radical addition” step is confusing: what is the source of the new carbonyl group? How does it be connected to III?).

Response: We appreciate the referee for giving this valuable point. The mechanism was concise in the beginning to meet the demand of the word and page limitation. We have provided more details regarding the mechanism in the manuscript which are marked with yellow color.

Comment 2. The yields and stereoselectivities (ee or dr) of all the corresponding products (Figures 1-10) should be added to give the readers more useful information.

Response: We appreciate the referee for this comment. We have added all relevant stereochemical information in the figures. Whenever no stereoselectivity was obtained, no asterisk was placed at the alpha-carbon center.

Comment 3. Page 11, “However, previous reports using Pd(0) as the catalyst together with achiral ligands suffered from affording the α -AAs with high enantioselectivities.”. This sentence is confusing. Why do high enantioselectivities make it suffer?

Response: We appreciate this valuable point from the reviewer. We have added the explanation in the manuscript as shown below:

However, previous reports using Pd(0) as the catalyst together with achiral ligands failed to afford the α -AAs with high enantioselectivities due to the distance between the stereogenic and catalytic centers.

Comment 4. One important work about synthesis of chiral α -AAs from amino acid Schiff bases should be cited: J. Am. Chem. Soc. 2017, 139, 29, 9819–9822.

Response: We appreciate this comment as this article was clearly missed in this contribution. This report by Zhang and co-workers has been added to the appropriate chapter.

Comment 5. Figure 10a, the equiv. of Cy2NMe is missed.

Response: We appreciate the referee for giving this point, we have added this information into the figure.

Comment 6. Ref. 62. The abbreviation of journal name is wrong.

Response: We appreciate the referee for this comment. We have corrected this mistake.

REVIEWERS' COMMENTS:

Reviewer #1 (Remarks to the Author):

This revised manuscript has made appropriate modifications based on referees' comments.

So now, this reviewer thinks this manuscript is suitable for the publication in Nature Communications.

Reviewer #2 (Remarks to the Author):

All the issues have been well addressed. Examples about catalytic asymmetric nucleophilic addition to ketimines have been added, making this review more comprehensive. Some ambiguous expressions are improved and the content is now more feasible to understand. And the related revision has been added carefully in the revised manuscript. Thus, this reviewer supports this manuscript to be published in Nature Communications in the present format.

Comments from reviewer 1 and our reply:

This revised manuscript has made appropriate modifications based on referees' comments. So now, this reviewer thinks this manuscript is suitable for the publication in Nature Communications.

Response: We appreciate the referee for accepting our modifications.

Comments from reviewer 2 and our reply:

All the issues have been well addressed. Examples about catalytic asymmetric nucleophilic addition to ketimines have been added, making this review more comprehensive. Some ambiguous expressions are improved and the content is now more feasible to understand. And the related revision has been added carefully in the revised manuscript. Thus, this reviewer supports this manuscript to be published in Nature Communications in the present format.